



# Forest harvesting impacts on micrometeorological conditions and sediment transport activities in a humid periglacial environment

Fumitoshi Imaizumi[1], Ryoko Nishii[2], Kenichi Ueno[3], Kousei Kurobe[4]

[1] Faculty of Agriculture, Shizuoka University, Shizuoka, 422-8529, Japan.
[2] Center for Transdisciplinary Research, Niigata University, Niigata, 950-2181, Japan
    [3] Faculty of Life and Environmental Sciences, University of Tsukuba, Tsukuba, Ibaraki, 305-8572, Japan.
    [4] Pacific Consultants Co., Ltd, Tokyo, 101-8462, Japan

*Correspondence to*: Fumitoshi Imaizumi (imaizumi@shizuoka.ac.jp)

**Abstract.** Sediment transport activities in the periglacial environment are controlled by hillslopes micrometeorological conditions (i.e., air and ground temperatures, ground water content), which are highly affected by vegetation cover. Thus, there is a possibility that forest harvesting, which is the most dramatic change to vegetation cover in mountain areas, may severely impact sediment transport activities in periglacial areas (i.e., soil creep, dry ravel). Knowledge of the effects of
forest harvesting on sediment transport are needed to protect aquatic ecosystems as well as to develop better mitigation measures for preventing sediment disasters. In this study, we investigated changes in sediment transport activities following forest harvesting in steep artificial forests located in a humid periglacial area of the Southern Japanese Alps. In the Southern Japanese Alps, rainfall is abundant in summer and autumn, and air temperatures frequently rise above and fall below 0 degrees in the winter. Our monitoring by time laps cameras revealed that gravitational transport processes (e.g., frost creep
and dry ravel) dominate during the freeze-thaw season, while rainfall-induced processes (surface erosion and soil creep) occur during heavy rainfall seasons. Removal of the forest canopy by forest harvesting alters the type of winter soil creep from deeper frost creep to diurnal needle-ice creep. Winter creep velocity of the ground surface sediment in the harvested site was significantly higher than that in the non-harvested site. Meanwhile, sediment flux on the hillslopes observed by sediment traps decreased in the harvested site. Branches of harvested trees left on the hillslopes captured sediment coming
from upslope. In addition, the growth of understories after harvesting possibly reduced surface erosion. Consequently, removal of the forest canopy by forest harvesting directly impacts micrometeorological conditions and periglacial sediment transport activity, while sediment flux on hillslopes is also affected by branches left on the hillslopes and recovery of understories.

## 1 Introduction

Vegetation cover is a factor that has large effects on micrometeorological conditions and sediment transport activities and can be easily controlled by humans (Sidle and Ochiai, 2006; Miyata et al., 2008). Forest harvesting is a common process that





causes large vegetation changes in mountain areas (Imaizumi et al., 2012; Ueno et al., 2015; Goetz et al., 2015). The impact of forest harvesting on the sediment transport process is possibly significant in periglacial areas, where sediment transport activities are highly controlled by micrometeorological conditions (e.g., air and ground temperatures, ground water conditions) (Matsuoka, 2001; Boelhouwers et al., 2000; Harris et al., 2008b). Therefore, forest harvesting impacts in

periglacial areas need to be understood to protect aquatic ecosystems as well as to develop better mitigation measures for preventing sediment disasters.

Previous studies emphasized that vegetation cover, especially a forest canopy, controls micrometeorological conditions on mountain hillslopes. Tree crowns intercept rainfall and reduce the volume of precipitation reaching the ground (Xiao et al., 2000; Fan et al., 2014). The tree crown also controls net radiation and ground temperature (Ueno et al., 2010; Ueno et al.,

2015). Sediment transport triggers, such as changes in soil moisture, generation of overland flow, and freeze-thaw of ground water, are affected by these micrometeorological conditions (Wainwright et al., 2000; Gray et al., 2002; Ueno et al., 2015). Therefore, removal of the canopy by forest harvesting alters sediment transport opportunities via changes to the micrometeorological conditions.

The sediment transport activity is also controlled physically by forest components (e.g., understory, tree roots, and woody

debris). The understory reduces kinetic energy of raindrops that splash soil particles, preventing surface erosion (Fukuyama et al., 2010; Nanko et al., 2015). Tree roots reinforce slope stability, reducing frequency of shallow landslides and debris flows (Imaizumi et al., 2008; Goetz et al., 2015). Woody debris captures sediment traveling on hillslopes (Hartanto et al., 2003; Imaizumi et al., 2017). Because forest harvesting dramatically changes these components, the sediment transport possibly change after the forest harvesting.

Mountain hillslopes are generally formed by a combination of various types of sediment transport processes (Roberts and Church, 1986; Benda, 1990; Imaizumi et al., 2017). However, most of the previous studies on the relationship between vegetation condition and sediment transport activities have been focused on a single sediment transport process (Miyata et al., 2009; Borrelli et al., 2015; Goetz et al., 2015). Sediment transport processes triggered by rainfall, such as surface erosion and landslides, are active in areas with abundant rainfall (Miyata et al., 2009; Fiorucci et al., 2011), while those triggered by

freeze-thaw are active in cold environments (Matsuoka, 2001; Boelhouwers et al., 2000; Boelhouwers et al., 2003; Harris et al., 2008b). In humid periglacial areas, both rainfall and freeze-thaw activities highly affect sediment transport processes (Imaizumi et al., 2015, 2017). Therefore, the effect of forest harvesting on sediment transport activities possibly changes depending on seasonal changes to the trigger mechanism and the predominant type of sediment transport.

The Southern Japanese Alps is characterized as a humid periglacial area because of abundant annual precipitation (>2500

mm) and frequent freeze-thaw cycles during winter, especially at mid elevation mountain ranges (e.g., 1000 to 2000 m)(Imaizumi et al., 2006; Imaizumi et al., 2017). Gravitational sediment transport processes (e.g., soil creep and dry ravel), which have been poorly studied in relation to vegetation change, are more important sediment transport processes than surface wash because of the steep terrain (Imaizumi et al., 2017). Ueno et al. (2015) showed that the type of winter solifluction changes from frost creep to needle-ice creep following forest harvesting in the Southern Japanese Alps, because



of changes in the net radiation and ground temperature. However, their interpretation of the sediment transport process is limited to winter frost heave and solifluction on a micro scale (e.g., < 1 m$^2$). Therefore, seasonal changes in the impact of forest harvesting on overall hydrogeomorphic processes have not been clarified.

The aim of this study is to clarify the impacts of forest harvesting on micrometeorological conditions and sediment transport processes throughout the year in a humid periglacial area. We studied the seasonal changes in the impact of forest harvesting on the micrometeorological conditions (i.e., radiation, ground temperature, throughfall, freeze-thaw of ground water) and sediment transport (i.e., soil creep, dry ravel, surface erosion) by intensive and comprehensive monitoring of harvested and non-harvested forests in the Southern Japanese Alps.

## 2 Study site

The study site was set in Ikawa University Forest, University of Tsukuba, located on the southern side of the Sothern Japanese Alps, central Japan (Fig. 1). The geology is composed of alternating sandstone and shale layers covered by brown forest soil and podzol. Annual precipitation is high (an annual average of 2800 mm from 1993 to 2002; Imaizumi et al., 2010) because the area lies in the East Asia Monsoon region. Heavy rainfall events (daily rainfall >100 mm day$^{-1}$) mainly occur in the early summer (June to July) due to a seasonal rain front, and in the typhoon season (August to early October).

Winter snowfall occurs from December to March, but precipitation in this period accounts for only about 15% of the total annual precipitation. Air temperatures frequently rise above and fall below 0 degrees during the winter (Ueno et al., 2015; Imaizumi et al., 2017).

The target study site was 1.5 ha and faced west at 1180–1310 m a.s.l. in the University Forest (Fig. 1). Japanese cypress (evergreen needle leaf) trees were planted over the study site in 1975. Annual herbaceous plants (e.g., Leuscsceptrum and

fern) covered 0 to 20 % of the ground surface before forest harvesting (Fig. 1c). A 0.87-ha area in the upper part of the study area, called clear-cut (CC) site in this study, was harvested during March–September 2012 (Ueno et al., 2015). Logging was conducted by skyline yarding to avoid damaging the slope surfaces by dragging logs. After the clear-cutting, branches of harvested trees were piled up on lines parallel to the counter lines with spacing of 5 to 10 m (Fig. 1b). This work is aimed to arrange the environment for revegetation and is common in Japan. After the clear-cutting, Japanese cypress trees were

replanted in spring 2013. Herbaceous plants (e.g., Japanese knotweed, artemisia, brambles, and thistle) covered more than 90% of the ground by July 2013 (Fig. 1d). No forest management was conducted in the lower part of the study site, which is called the non-cutting (NC) site. The slope gradient at site CC (35–45°) was slightly greater than that at site NC (30–40°).

We set up three monitoring plots corresponding to the small-scale topography in the CC site (CCR, CCS, and CCV at ridge, straight, and valley shaped cross-sectional topographies, respectively) and two monitoring plots in the NC site (NCR

and NCS at ridge and straight shaped cross-sectional topographies, respectively (Fig. 1, Table 1). Contributing area of plots at ridge shaped slopes (CCR and NCR) was smallest in size and that at a valley shaped slope (CCV) was the largest in the



size. Grain size at plot CCV was also clearly higher than at the other plots due to the accumulation of boulders transported as rockfall and dry ravel from the upper slopes (Fig. 2).

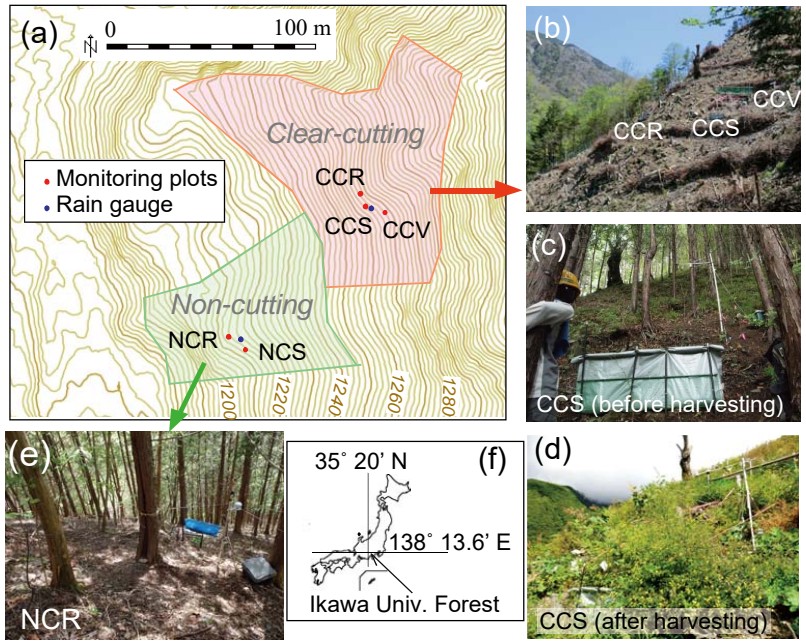

**Figure 1: Topographic map and photographs showing observation sites in the Ikawa University Forest. (a) Topographic map of the study site. (b) Photograph of site CC after forest harvesting (May 13, 2013). (c) Photograph of plot CCS before forest harvesting (August 26, 2011). The synthetic sheet at the lower end is the sediment trap. A radiometer was attached on the horizontal metal pipe extending from a white vertical pole in the right side of the image. (d) Photograph of plot CCS after forest harvesting (September 26, 2013), taken at the same location as Fig. 1c. (e) Photograph of plot NCR (May 14, 2015). (f) Location of the Ikawa University Forest within Japan.**

**Table 1: Setting of monitoring plots.**

| Plot | Cross-sectional topography | Slope gradient[a] (degree) | Contributing area (m$^2$) |
|------|----------------------------|-----------------------------|----------------------------|
| CCR | ridge | 43 | 7 |
| CCS | straight | 43 | 28 |
| CCV | valley | 40 | 70 |
| NCR | ridge | 34 | 6 |
| NCS | straight | 38 | 40 |

[a] slope gradient measured 10 m upslope from sediment traps





### 3 Methodology

#### 3.1 Meteorological and freezing-thawing condition

Preliminary observation of the micrometeorology, including throughfall and ground surface temperature, was carried out on both the CC and NC sites in the period from June 2011 to March 2012, before forest harvesting (periods 1 to 11 in Table 2). The observation was continued in site NC during the harvesting period (periods 12 to 14), while that was intermitted in site CC. After the forest harvesting, intensive monitoring of radiation budget, snow depth, and ground temperature was conducted in the periods from January 28 to May 27, 2013 and from December 2, 2013 to March 24, 2014 in both sites (Tables 2, 3).

Rainfall (throughfall) at both sites (CC and NC) was observed using tipping bucket rain gauges (0.2 mm for one tip) without a heating system. Rainfall monitoring in site NC was intermitted for more than several months due to troubles with the monitoring system (bad electrical connections, choking of the rain gauge by leaf fall, troubles with the data logger) in the summers of 2013 and 2014. In addition, there were some other short intermissions (duration < 1 month) in both sites CC and NC due to mechanical troubles. Gross rainfall with 1-minute interval was monitored at site MU (1060 m a.s.l.), located 1 km south west of the study site.

To evaluate the precise radiation budget at the surface, four radiation components were measured by radiometers, which were attached to the horizontal metal pipe 2.0 m above the ground (Fig. 1c). The ground surface temperature was observed by temperature data loggers, which were covered by small cobbles to prevent direct hit of the solar insolation. The temperature observed by the temperature data loggers has a lower spatial representation than that estimated from upward longwave radiation by the Stefan–Boltzmann law. However, we used the monitoring data from the temperature data logger, because the duration of the monitoring period of the radiometer was short. Ground temperatures at 0.05, 0.15, and 0.30 m depths were monitored by thermocouples. Snow depth was observed by ultrasonic sensors, which were installed on poles at heights of about 1.7 m and measured the distance between the snow surface (ground surface in snow-free periods) and the sensors. Air temperature was observed by thermo-hygrographs.



**Table 2: Observation periods of sediment traps (periods 1 to 27) and operation periods of other monitoring devises. R, TLC, and RG indicates radiometers, time lapse cameras, and rain gauges, respectively. Monitoring of sediment traps in site CC was intermitted during the harvesting period. Monitoring periods of ultrasonic sensors for snow depth, thermocouples, and extensometers were the same as that of radiometers.**

| Periods | Start date | End date | Duration (day) | Season | Other monitoring device | | | | |
|---|---|---|---|---|---|---|---|---|---|
| | | | | | R | TLC (CC) | TLC (NC) | RG (CC) | RG (NC) |
| 1 | Jun. 22, 2011 | Jul. 5, 2011 | 13 | Rainfall | | | | | O |
| 2 | Jul. 5, 2011 | Aug. 23, 2011 | 49 | Rainfall | | | | | O |
| 3 | Aug. 23, 2011 | Oct. 7, 2011 | 37 | Rainfall | | | | O | |
| 4 | Oct. 7, 2011 | Nov. 4, 2011 | 36 | Rainfall | | | | O | O |
| 5 | Nov. 4, 2011 | Nov. 25, 2011 | 21 | Rainfall | | | | | O |
| 6 | Nov. 25, 2011 | Dec. 13, 2011 | 18 | FT | | | | O | O |
| 7 | Dec. 13, 2011 | Jan. 24, 2012 | 42 | FT | | | | O | O |
| 8 | Jan. 24, 2012 | Feb. 2, 2012 | 9 | FT | O | | | O | O |
| 9 | Feb. 2, 2012 | Feb. 20, 2012 | 18 | FT | O | | | O | O |
| 10 | Feb. 20, 2012 | Mar. 6, 2012 | 15 | FT | O | | | O | O |
| 11 | Mar. 6, 2012 | May 8, 2012 | 63 | FT and Rainfall | O | | | O | O |
| 12 | May 8, 2012 | Aug. 10, 2012 | 94 | Rainfall | *Harvesting period* | | | | O |
| 13 | Aug. 10, 2012 | Sep. 25, 2012 | 46 | Rainfall | *Harvesting period* | | | | |
| 14 | Sep. 25, 2012 | Oct. 2, 2012 | 7 | Rainfall | *Harvesting period* | | | | |
| 15 | Oct. 2, 2012 | Nov. 19, 2012 | 48 | Rainfall | | | | | |
| 16 | Nov. 19, 2012 | Dec. 27, 2012 | 38 | FT | O | O | O | O | O |
| 17 | Dec. 27, 2012 | Jan. 28, 2013 | 32 | FT | O | O | O | O | O |
| 18 | Jan. 28, 2013 | Mar. 4, 2013 | 35 | FT | O | O | O | O | O |
| 19 | Mar. 4, 2013 | Mar. 21, 2013 | 17 | Rainfall | O | O | O | O | O |
| 20[*2] | Mar. 21, 2013 | May 27, 2013 | 67 | Rainfall | | O | O | O | O |
| 21 | May 27, 2013 | Aug. 20, 2013 | 85 | Rainfall | | O | | O | O |
| 22 | Aug. 20, 2013 | Nov. 12, 2013 | 84 | Rainfall | | | | O | |
| 23 | Nov. 12, 2013 | Jan. 17, 2014 | 66 | FT | | O | | O | O |
| 24 | Jan. 17, 2014 | Apr. 4, 2014 | 77 | FT | | O | | O | O |
| 25 | Apr. 4, 2014 | Jul. 29, 2014 | 116 | Rainfall | | | | O | O |
| 26 | Jul. 29, 2014 | Nov. 11, 2014 | 105 | Rainfall | | | O | O | |
| 27 | Nov. 11, 2014 | May 25, 2015 | 195 | FT | | | | O | |



**Table 3: List of monitoring devices**

| Monitoring item | Devices | Model (manufacture) | Interval (minute) | Accuracy | Location |
|---|---|---|---|---|---|
| Rainfall | Tipping bucket rain gauges | Rain collector II (Davis instruments) | 1 | 4% | CC, NC (Fig. 1) |
| Four radiation components | Net Radiometers | CNR 4 (Kipp & Zonen Co.) | 10 | 4% | CCS, NCS |
| Air temperature | Thermo-hygrographs | Hobo Pro v2 U23(Onset Co.) | 10 | 0.2 °C | CCS, NCS |
| Ground surface temperature | Temperature data loggers | TidbiT v2 (Onset Co.) | 10 | 0.2°C | All plots |
| Ground temperature | Thermocouples | - | 10 | 1 °C | CCR, NCR |
| Snow depth | Ultrasonic sensors | U-GAGE T30UXUB (Banner Engineering Corp.) | 10 | 0.25% | CCR, NCR |
| Frost heave | Extensometers | DT-100A (Kyo-WA Co.) | 60 | 0.5% | CCR, NCR |
| Frost heave, Soil creep | Time lapse camera | GardenWatchCam (Brinno) | 10 | - | CCR, NCR |

### 3.2 Frost heave and soil creep velocity

The occurrence of frost heave was detected by extensometers and TLCs (Ueno et al., 2015). The main body of the
extensometer was fixed to a metal beam located 0.4 m above the ground surface. The edge of the detection part extending
down from the bottom of the main body was attached to the ground surface. The output voltage was converted to
displacement (along a line running perpendicular to the ground surface) using a calibration formula for each sensor. The
biases caused by the changes in the temperature in the non-freezing periods were corrected using air temperature monitored
at both sites. However, a maximum bias, due to changes in the air temperature, of 5.0 mm remained in site CC following
correction equations as a result of large diurnal changes in air temperature.

Time laps cameras (TLCs) were set 0.2 m above the ground after forest harvesting (in November 2012), and shot ground
surface around the extensometers in the daytime. Temporal changes in the ground surface level were interpreted by the
image analyses using scales placed on the ground surface. Frost heave values observed by the extensometers were generally
lower than those from the camera images because of penetration of the head of the detection part into the loose ground
surface. Therefore, we converted the displacement values recorded by the extensometers to actual displacement by applying
the calculated relationship between the displacement recorded by the extensometers and that from the camera images.
Velocity of the ground surface sediment in the slope direction associated with the soil creep was estimated by comparing the
location of at least three pebbles on the ground surface in images. In site CC, images in the period from June 23 to December
1, 2012, and after March 5 could not be used for analysis of frost heave and soil creep, because the ground surface sediment
was completely covered by living and dead grasses. Data at site NC is not available in the period from May 11 to July 29,
2014, due to TLC mechanical trouble and loss of data due to a crash of the digital storage. Because of limitations in the



resolution of camera images (about 1 to 1.5 mm/pixel in the study area), we were only able to identify frost heave and soil creep with displacement > 1–2 mm.

### 3.3 Sediment flux

Sediment traps were installed at all five monitoring plots to record the sediment flux by ground surface and near-surface processes, such as soil creep, dry ravel, rockfall, and surface erosion. The locations of all sediment traps in site CC were over 5 m away from branches piled up by the forestry operation, because sediment flux just below the piled branches was likely to be lower than the surroundings. Vertical wire meshes (1.75 m wide) were secured on the hillslope using steel bars (Figs. 1c, 3). Synthetic sheets were placed on the upslope side of the wire mesh and adjacent ground surface to facilitate capture of finer sediments (e.g., sand and silt), as well as to distinguish between residual soil and sediment transported from the upper slopes. The sides of the synthetic sheets adjacent to the ground surface were closed to prevent removal of fine sediment in the trap by rainfall and surface water (Fig. 3). Since the synthetic sheets were fixed on the ground surface, sediment traps did not measure soil creep in the subsoil (>0.05 m in depth).

Sediment stored in the traps was collected 27 times between July 5, 2011 and May 25, 2015 (Table 2). Sampling intervals varied from 7 to 195 days, but generally we attempted to capture the major periods of potential seasonal differences (i.e., periods of the seasonal rainfall front, typhoon seasons, and extreme periods of freezing and thawing) that would affect the type of sediment transport. We classified the 27 sampling periods into rainfall seasons (basically from April to November) and freeze-thaw seasons (basically from December to March) in order to investigate seasonal differences in the sediment transport characteristics (Table 2).

Sediment traps aimed to observe the sediment flux from the contribution area over a hillslope scale. Therefore, sediment flux observed by sediment traps can be affected by amount and spatial distribution of branches left in the harvested area. In contrast, previously explained TLCs aimed to monitor soil creep within small scale target areas (< 1 m$^2$).

Sediment larger than 30 mm stored by the traps was weighed in the field using a spring balance. Sediment smaller than 30 mm were taken to the laboratory. After drying in a drying oven at a temperature of 105ºC for about eight hours, large organic materials (>4 mm) were removed by hand. Then, grain-size distribution was analyzed using sieves with mesh sizes of 4, 8, and 16 mm.



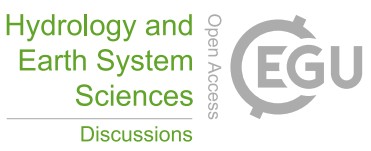

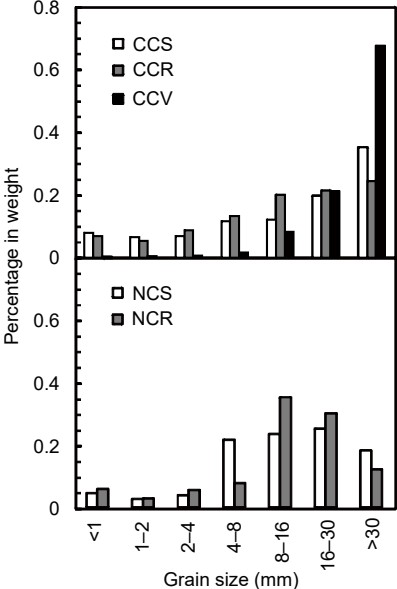

**Figure 2: Grain size distribution of the ground surface sediment around monitoring plots.**

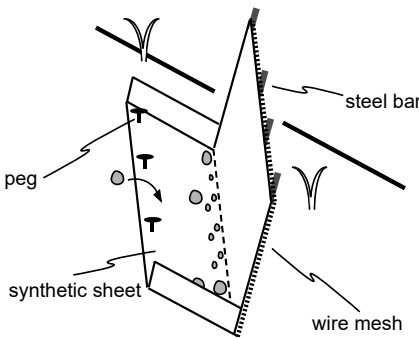

peg

steel bar

synthetic sheet

wire mesh

**Figure 3: Illustration of sediment traps.**

**4 Results**

**4.1 Micrometeorological and freezing-thawing conditions**

Clear changes in the winter micrometeorological conditions, including ground surface temperature, net radiation, and snow depth, following forest harvesting were monitored in site CC (Figs 4, 5). Although the ground surface temperature at CC was





similar to NC before forest harvesting, diurnal temperature range increased following harvesting. Ground surface temperature in CCS was 2.6 ºC lower than that in NCR at 7:00 am and 14.6 ºC higher at 14:00 pm (Fig. 4). Net radiation in site CC is lower than that in site NC in the morning, and is notably higher than that of NC around noon (Fig. 5e). This is because of high upward longwave radiation in the morning and downward shortwave radiation around noon in CC (Figs. 5c, 5d).

Large diurnal changes in the ground temperature in CC (18.4 ºC in Fig. 4) resulted in a high frequency of freeze-thaw cycles at the ground surface (Fig. 5f). In site CC, more than 10 diurnal freeze-thaw cycles were observed at 0.05 m depth, whereas just several were observed at a depth of 0.15 m (Fig. 5g). Ground temperature did not drop below 0 ºC at 0.30 m depth. Freeze-thaw cycles at the ground surface and 0.05 m depth in site NC are characterized by a low frequency and long

duration (longer than a week, Figs. 5f, 5h). Ground temperature at 0.30 m depth in site NC was also below 0 ºC for several days during the long periods of snow cover (Figs. 5b, 5h). The diurnal freeze-thaw cycle frequently occurred in site CC until the end of March, while ground temperature seldom fell below 0 degrees in site NC in March (Fig. 6).

Snow depth and snow cover duration were also different between sites CC and NC. Snow depth at site CC, where the forest canopy which intercepted snow was removed by the harvesting, was higher than that at NC following heavy snow fall events

(e.g., January 15, 2012, Fig. 5b). At the same time, duration of the snow cover in site CC was shorter than that in NC because the snow depth decreases at a higher rate in site CC than site NC.

In rainfall seasons, changes in the micrometeorological condition following forest harvesting was evident in the throughfall amount (Fig. 7). Differences in the hourly rainfall intensity between sites CC and NC was not clear before forest harvesting (Fig. 7a), while the rainfall intensity in CC was 0 to 3 mm hr$^{-1}$ higher than that in NC after forest harvesting (Fig.

7b).

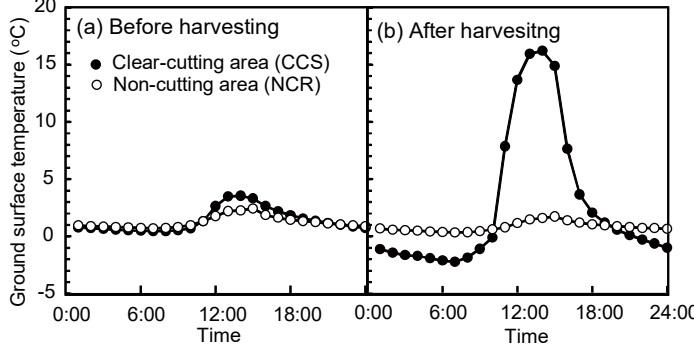

**Figure 4: Average ground surface temperature before and after harvesting at site CC (CCS) and NC (NCR) observed by temperature data loggers. (a) Before clear cutting in the period from December 1, 2011 to February 29, 2012. (b) After clear cutting in the period from December 1, 2012 to February 28, 2013.**



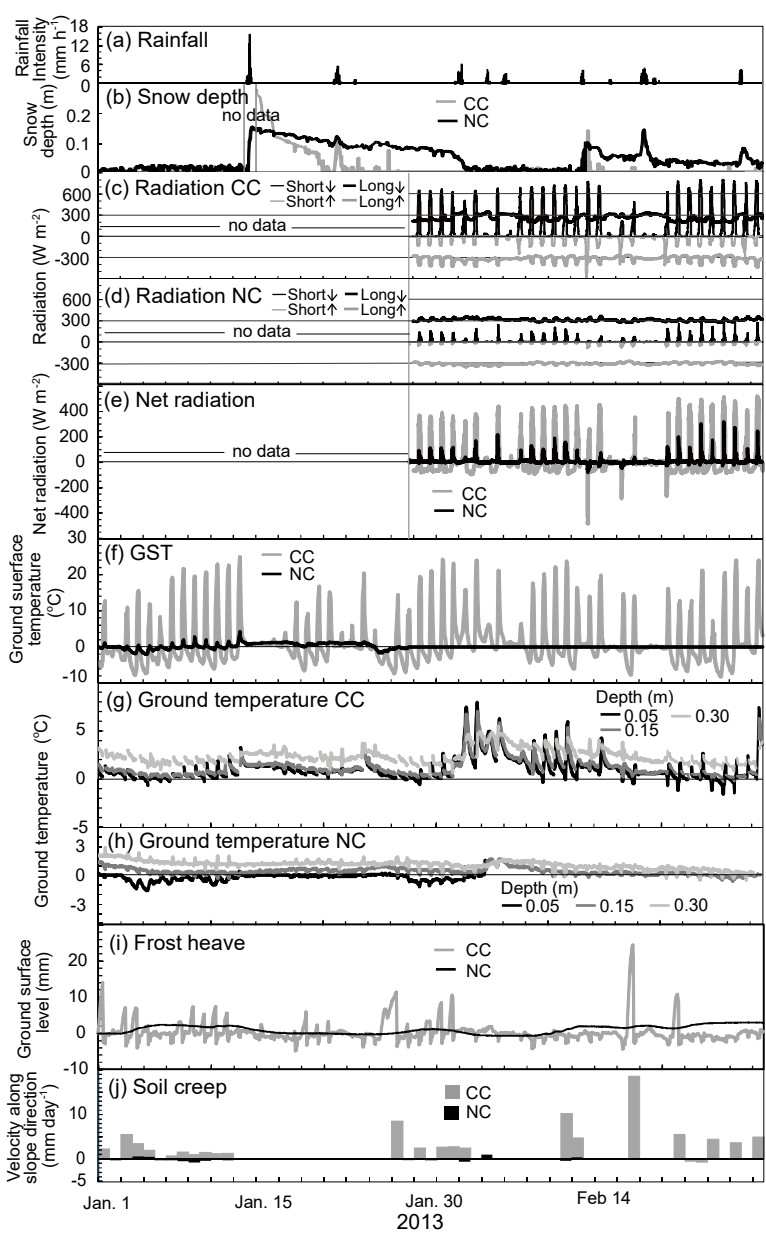





Figure 5: Comparison of micrometeorological conditions and sediment transport activities between sites CC and NC in winter (January 1 to February 28, 2013). (a) Rainfall intensity at site MU. (b) Snow depth measured by ultrasonic sensors at sites CC (CCR) and NC (NCR). (c) Radiation measured by a net radiometer in site CC (CCS). (d) Radiation measured by a net radiometer in site NC (NCS). (e) Net radiation measured by net radiometers in sites CC (CCS) and NC (NCS). (f) Ground surface temperature (GST) in sites CC (CCR) and NC (COR) measured by temperature loggers. (g) Ground temperature in CC (CCR) measured by thermocouples. (h) Ground temperature in NC (NCR) measured by thermocouples. Monitoring of the ground temperature at 5 cm depth was interrupted by the defect of the thermocouple from February 4. (i) Changes in the ground surface level measured by extensometers at CC (CCR) and NC (NCR). (j) Velocity of ground surface sediment along slope direction obtained from TLC images.

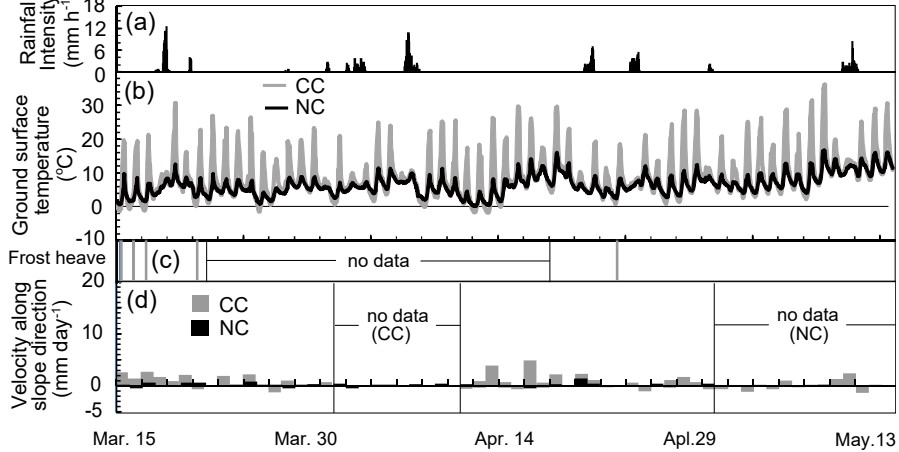

Figure 6: Comparison of micrometeorological conditions and sediment transport activities between sites CC and NC in spring (March 15 to May 15, 2013). (a) Hourly rainfall intensity at site MU. (b) Ground surface temperature in sites CC (CCR) and NC (NCR). (c) Timing of frost heave in CCR monitored by TLS. Uplifting of the ground surface was not observed at NC in this period. (d) Velocity of ground surface sediment by soil creep in CC (plot CCR) and NC (plot NCR) obtained from TLS images.





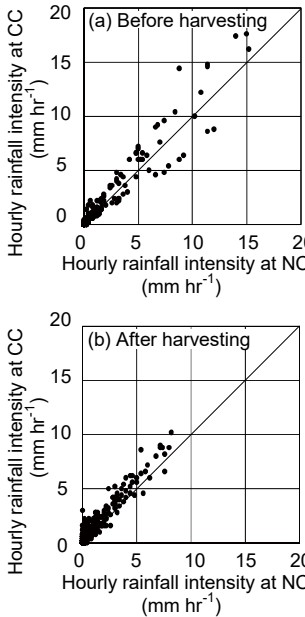

**Figure 7: Comparison of the hourly rainfall intensity between sites NC and CC. (a) Comparison of before forest harvesting (from August 25 to September 2 and from November 4 to November 30, 2012). The rainfall monitoring was intermitted by the damage of the rain gauge associated with a Typhoon (September 2 to October 7) and by a malfunction of the data logger (October 7 to November 3). (b) Comparison of after forest harvesting (April 4 to July 3, 2012).**

5 ## 4.2 Frost heave and soil creep

In winter, ground surface level changes frequently due to frost heave in site CC, except during periods of snow cover, while frost heave in site NC has a longer cycle (Fig. 5i). Uplifting period of the ground surface in site NC roughly corresponded to the freezing period of the subsoil (i.e., 0.05 m depth). Velocity of the ground surface sediment by soil creep observed by a TLC in site CC was generally high (>2 mm day$^{-1}$) on the days with frost heave (Fig. 5i, 5j), which was

10 significantly higher than that in site NC.

In site CC, frost heave was less frequent in the early spring (March), with almost no frost heave occurring in late spring (late April to March; Fig. 6). Frost heave in this period was only monitored by the TLC, as frost heave height in March was lower than the bias of extensometers (≤ 5 mm). No changes in the ground surface level were identified in site NC. Displacement of ground surface sediment in site CC still occurred frequently until the end of April. High velocity (> 2 mm

15 day$^{-1}$) was not only observed on frost-heave days but also on other days, including those with no precipitation. Velocity of the ground surface sediment in site NC during the spring periods was much lower than that in site CC (Fig. 6).




During the rainfall seasons, velocity of the ground surface sediment was high at both sites CC and NC (> 2 mm day$^{-1}$), on days with heavy rainfall events (Fig. 8). The rainfall threshold for such high velocities was roughly given by a maximum hourly rainfall intensity of > 5 mm h$^{-1}$ and a total rainfall depth of > 40 mm for both sites CC and NC (Fig. 9).

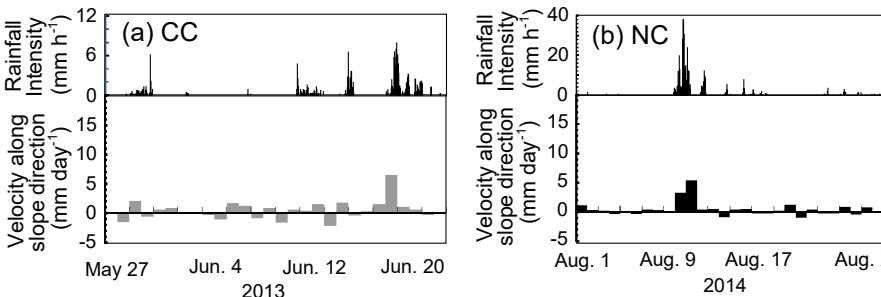

**Figure 8: Velocity of ground surface sediment during rainfall seasons. (a) Velocity at site CC (CCR) in the period from May 27 to June 23, 2013. (b) Velocity in site NC (NCR) in the period from August 1 to August 31, 2014.**

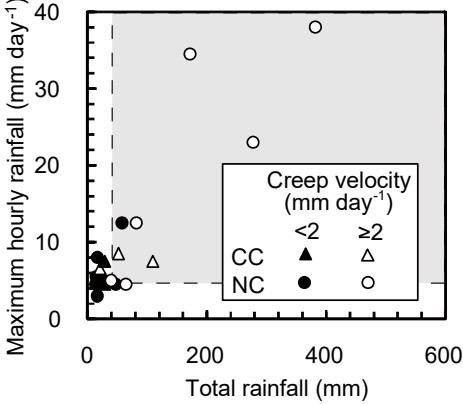

**Figure 9: Comparison between total rainfall depth and maximum hourly rainfall during rainfall events with and without clear displacement of ground surface sediment (velocity of <2 and ≥2 mm day$^{-1}$, respectively). Rainfall was monitored at MU. Shaded area indicates total rainfall depth of > 40 mm and hourly rainfall intensity of > 5 mm h$^{-1}$.**

### 4.3 Sediment flux

Sediment flux before forest harvesting was spatially variable among monitoring plots (Fig. 10, Table 4). During the monitoring period, sediment flux at plot CCV, where sediment from surrounding slopes accumulates, was notably higher than that at the other monitoring plots (Table 4). Sediment flux in the winter (from Nov. 25, 2011 to March 6, 2012) was clearly higher than that in the rainfall seasons (June 22 to November 24, 2011) at three plots (CCR, CCV, NCR; Table 4). In





addition, if two exceptionally large boulders (> 1 kg), which occupied 50.0 % of the sediment flux in the rainfall season were excluded, then sediment flux in the winter at NCS (11.1 kg m$^{-1}$ day$^{-1}$) was higher than that in the rainfall season (8.9 kg m$^{-1}$ day$^{-1}$). Coarse sediment (>16 mm) occupies a large portion of the sediment flux in both rainfall and freeze-thaw seasons at all observation plots except at CCR during rainfall seasons (Fig. 11). Sediment flux of coarse sediment (>16 mm) in freeze-

5 thaw seasons was higher than that in rainfall seasons at all plots (Fig. 11). In contrast, sediment flux of fine sediment (<4 mm) during rainfall seasons is higher than that in freeze-thaw seasons at all plots in CC before forest harvesting (Figs. 11a–11f) and at all plots in NC (Fig. 11g–11j).

Sediment flux in site CC after the 2012 forest harvesting was lower than that before the harvesting both during rainfall and freeze-thaw seasons (Fig. 10, Table 4). Decreases in the sediment flux of all grain size classes was identified (Figs. 11a–11f).

10 In contrast, changes in the sediment flux before and after the harvesting period were not clear in site NC (Fig. 10, Table 4). The sediment flux of fine sediment during rainfall seasons was lower than that in freeze-thaw seasons at all of the plots in site CC after forest harvesting, despite the sediment flux of fine sediment being higher during rainfall seasons than freeze-thaw seasons before harvesting (Fig. 11g–11j).

Sediment weight in the artificial forest (in CC before the forest harvesting, and in NC) captured by sediment traps does not

15 have a clear relationship with total rainfall depth and maximum daily rainfall in the sampling period of sediment traps, except for CCS (Fig. 12, Table 5). Sediment weight after the harvesting in CC does not have a clear relationship with the rainfall factors either. In site CC, sediment weight after harvesting is less than that before harvesting, when compared with periods of similar rainfall depth and intensity.





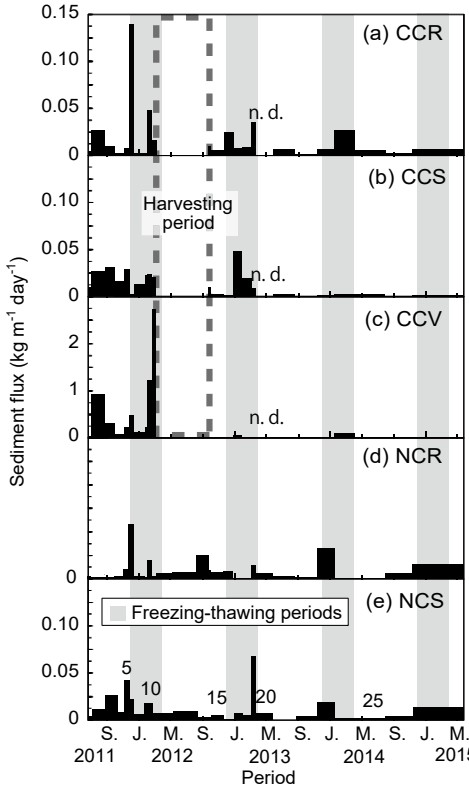

**Figure 10: Temporal changes in the sediment flux: (a) plot CCR, (b) CCS, (c) CCV, (d) NCR, and (e) NCS. Numbers in (e) indicates sampling periods listed in Table 1.**

**Table 4: Comparison of average sediment flux in rainfall and freeze-thaw (FT) seasons before and after forest harvesting. Values**
5 **in parentheses at plot NCS were calculated by excluding the weight of two exceptionally large boulders with weights of > 1 kg.**

| Plots | Sediment flux (x $10^{-3}$ kg m$^{-1}$ day$^{-1}$) | | | | C/A | D/B |
|---|---|---|---|---|---|---|
| | Before clearcutting | | After clearcutting | | | |
| | Rainfall season (A) | FT season (B) | Rainfall season (C) | FT season (D) | | |
| CCR | 12.1 | 35.5 | 3.6 | 12.1 | 0.29 | 0.34 |
| CCS | 21.8 | 14.4 | 1.1 | 6.0 | 0.05 | 0.42 |
| CCV | 418.1 | 753.9 | 5.4 | 20.1 | 0.01 | 0.03 |
| NCR | 2.5 | 14.5 | 3.2 | 12.3 | 1.26 | 0.85 |
| NCS | 17.8 (8.9) | 11.1 | 3.1 | 11.1 | 0.17 (0.34) | 1.01 |



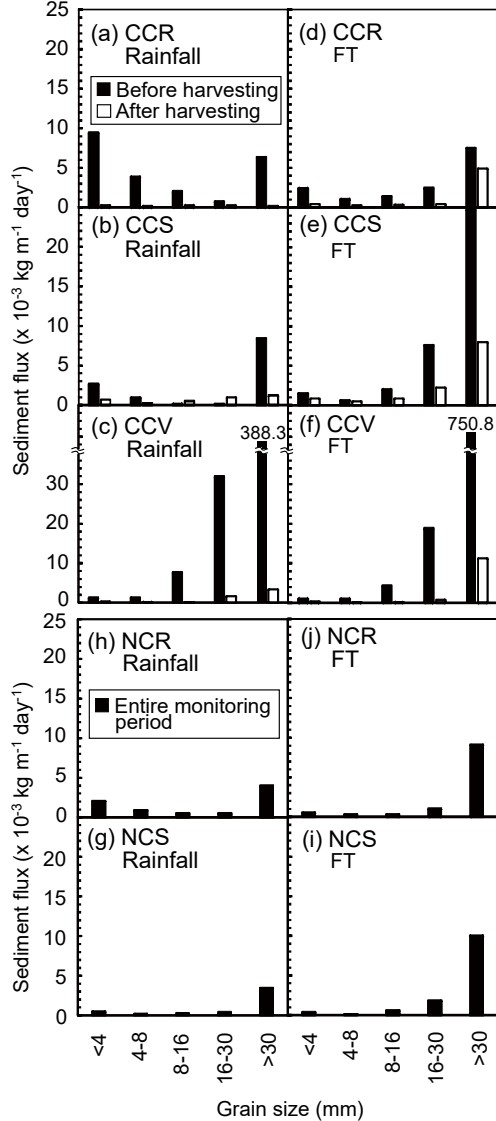

**Figure 11: Average sediment flux of each grain size class in harvested area (site CC). (a) Sediment flux at plot CCR in rainfall periods. (b) Sediment flux at plot CCS in rainfall periods. (c)Sediment flux at plot CCV in rainfall periods. (d) Sediment flux at plot CCR in freeze-thaw periods. (e) Sediment flux at plot CCS in freeze-thaw periods. (f) Sediment flux at plot CCV in freeze-thaw periods. (g) Sediment flux at plot COR in rainfall periods. (h) Sediment flux at plot COS in rainfall periods. (i) Sediment flux at plot COR in freeze-thaw periods. (j) Sediment flux at plot COS in freeze-thaw periods.**





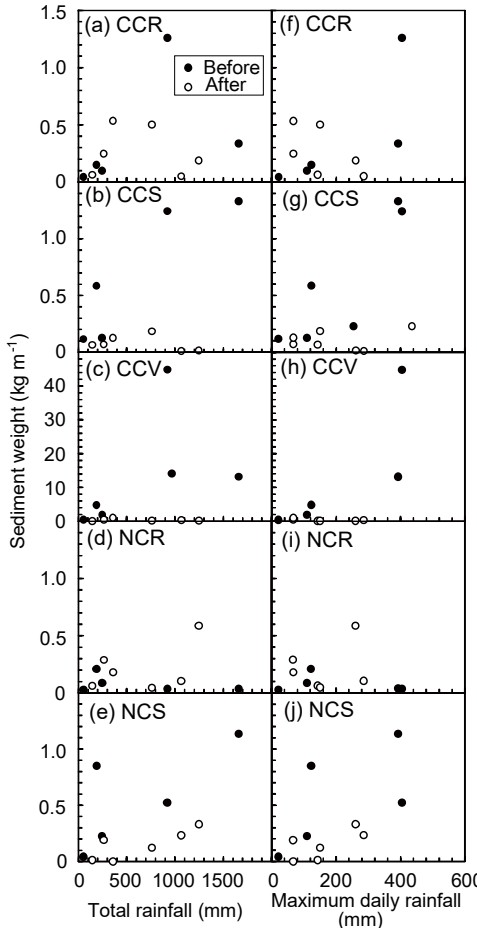

**Figure 12: Comparison between rainfall factors and sediment weight captured by sediment traps during rainfall seasons. (a) Total rainfall and sediment weight at plot CCR. (b) Total rainfall and sediment weight at plot CCS. (c) Total rainfall and sediment weight at plot CCV. (d) Total rainfall and sediment weight at plot NCR. (e) Total rainfall and sediment weight at plot NCS. (f) Maximum daily rainfall and sediment weight at plot CCR. (g) Maximum daily rainfall and sediment weight at plot CCS. (h)**
5 **Maximum daily rainfall and sediment weight at plot CCV. (i) Maximum daily rainfall and sediment weight at plot NCR. (j) Maximum daily rainfall and sediment weight at plot NCS.**





**Table 5: R-squared and p values of linear regressions between sediment weight captured by sediment traps and rainfall factors in rainfall seasons.**

| Plot | Before harvesting | | | | After harvesting | | | |
|---|---|---|---|---|---|---|---|---|
| | Total rainfall (mm) | | Maximum daily Rainfall (mm/day) | | Total rainfall (mm) | | Maximum daily rainfall (mm/day) | |
| | $R^2$ | P | $R^2$ | P | $R^2$ | P | $R^2$ | P |
| CCR | 0.20 | 0.45 | 0.60 | 0.13 | 0.03 | 0.75 | 0.34 | 0.23 |
| CCS | 0.80 | 0.04 | 0.93 | 0.01 | 0.14 | 0.46 | 0.38 | 0.19 |
| CCV | 0.24 | 0.40 | 0.64 | 0.10 | 0.08 | 0.59 | 0.27 | 0.29 |
| NCR | 0.17 | 0.49 | 0.12 | 0.58 | 0.12 | 0.47 | 0.01 | 0.83 |
| NCS | 0.53 | 0.16 | 0.42 | 0.24 | 0.16 | 0.41 | 0.04 | 0.67 |

## 5 Discussion

### 5.1 Sediment transport characteristics in steep artificial conifer forest

Sediment transport activities were observed during both rainfall and freeze-thaw seasons in the conifer artificial forest NC (Fig. 10). As reported by Ueno et al. (2015), long frost heave cycles (> 1 week) and thick freezing layers (0.15–0.30 m deep in some periods) indicate that frost creep occurs in site NC (Figs. 5h, 5i). In contrast, even though there is same elevation zone, diurnal frost heave and needle ice creep dominate in the deciduous broadleaf forest located in the Ikawa University forest, in which controls of radiation, ground temperature, and soil moisture by the forest canopy are limited in the winter
season (Imaizumi et al, 2017). Therefore, the type of periglacial sediment transport process differs depending on the dominant tree species.

Strong rainfall events during rainfall seasons also caused soil creep in site NC (Figs. 8, 9). However, rainfall factors did not have a clear relationship with sediment transport activities (Fig. 12). In addition, the periods with the highest sediment flux differed among the plots (Fig. 10). These are likely to be affected by episodic sediment supply, such as small slope
failure and the release of sediment from woody debris (e.g., Kirchner et al., 2001; Imaizumi et al., 2015). In addition, some sediment transport processes (e.g., dry ravel and rockfall) are triggered not only by the rainfall, but also by other mechanisms (e.g., decrease in cohesion by evaporation of soil moisture, wind, and disturbance by animals) (Verity and Anderson, 1990; Gabet, 2003), also obscuring the relationship between rainfall factors and sediment transport activities.

Because the ratio of coarse sediment (>16 mm) in all sediment captured by sediment traps was significantly higher than
the ratio of coarse sediment in the ground surface sediment (Figs. 2, 11), it is likely that coarse sediment was selectively transported on the hillslope. The travel distance of coarse sediment as rockfall and dry ravel is longer than that of fine sediment (Dorren, 2003; Haas et al., 2012). Therefore, rockfall and dry ravel, which are active on slopes with a similar or steeper slope gradient than the angle of repose (Gabet, 2003; Lamb et al., 2011), are the most important transport types in the steep Ikawa University Forest. The ratio of fine sediment, which is selectively transported by overland flow (Heng et al.,



2011; Zhao et al., 2014), in the entire sediment flux was less than its ratio in the ground surface sediment (Figs. 2, 11). Therefore, surface erosion is less important than gravitational sediment transport.

Sediment flux was spatially different depending on the slope shape (Fig. 10). Sediment flux on the valley shaped slope, which had the largest contributing area, was higher than the straight and ridge shaped slopes in CC before forest harvesting
(Fig. 10). A similar trend was also observed in the natural deciduous forest in the Ikawa University Forest (Imaizumi et al., 2017). The contributing area may affect rapid sediment transport with long travel distances (i.e., dry ravel and rock fall) rather than slow sediment transport (i.e., soil creep), which can be explained by local freezing and transport conditions without any consideration of the contributing area (Higashi and Corte, 1971; Matsuoka, 1998). Before forest harvesting, the ratio of coarse sediment in CCV was higher than CCS and CCR. Therefore, high rockfall and dry ravel activity in the valley-
shaped slope may have increased the sediment flux of coarse sediment (Fig. 2). Because fine sediment is needed for the development of frost (Boelhouwers, 1998), soil creep was likely less active at the plot CCV because of a lack of fine sediment on the ground surface (Fig. 2).

### 5.2 Forest harvesting impacts on micrometeorological conditions

Ground temperature was one of the micrometeorological conditions most affected by forest harvesting (Figs. 4, 5f).
Frequency of the freeze-thaw cycle in site CC after forest harvesting (e.g. 50 times in the period from January 1 to February 28, 2013) was higher than that in site NC (18 times in the same period) because of the large diurnal changes in the ground temperature (Fig. 4, 5f). This is basically due to increases in the downward shortwave radiation during daytime and increases in the upward longwave radiation in the morning due to the removal of the tree crown (Figs. 5c, 5d; Ueno et al., 2015). Deep seasonal freeze-thaw in the conifer artificial forest (depth of >0.15 m) changed to shallow diurnal freeze-thaw (depth <0.05
m) after forest harvesting (Fig. 5f, Fig. 13).

Snow depth and duration of snow cover were also changed by forest harvesting (Fig. 5b). Snow depth after snowfall events in CC was higher than in NC, because of the loss of canopy interception. At the same time, duration of snow cover in CC was significantly shorter than NC because of the higher downward radiation to the snow surface and higher daytime temperatures (Figs. 5c, 5f). Such a short duration of the snow cover also facilitated diurnal changes in the ground
temperature, and increased the frequency of freeze-thaw cycles. Snow cover also affects soil moisture, which control frost heave and soil creep activities (Meentemeyer and Zippin 1981; Matsuoka 2001; Boelhouwers et al., 2003; Blankinship et al., 2014). Daytime soil moisture was generally lower during snow free periods than snow covered periods due to evaporation by insolation (Blankinship et al., 2014; Ueno et al., 2015).

Rainfall (throughfall) intensity in site CC was higher than that in NC after forest harvesting (Fig. 7). Loss of canopy
interception by forest harvesting may have increased throughfall in site CC (Xiao et al., 2000; Fan et al., 2014).





### 5.3 Forest harvesting impact on soil creep activity

Impact of forest harvesting on soil creep activities monitored by TLCs was most evident in winter and spring, when freeze-thaw of the groundwater directly or indirectly triggered soil creep (Figs. 5, 6). In the winter, large diurnal changes in ground temperature facilitated frequent frost heave in CC, resulting in the high velocity of soil creep (Fig 5i, 5j)(Ueno et al., 2015). In contrast, frost heave activities in NC can be characterized as having long cycles and low frequencies due to small diurnal changes in the ground temperature. Therefore, forest harvesting changed the type of sediment transport from seasonal frost creep, which is common in seasonal freeze-thaw areas with a thick freezing depth, to needle ice creep, which is observed in areas with shallow diurnal freeze-thaw activity (Fig. 13)(Boelhouwers, 1998; Matsuoka, 2001; Imaizumi et al., 2015). Thickness of soil creep layers, which are affected by the thickness of the freezing layer (Matsuoka, 2001; Harris et al., 2008a), were larger in site NC than in site CC. Therefore, a high velocity of ground surface sediment in site CC does not simply result in increases in the sediment transport rate.

In site CC, soil creep velocity in early spring (from late March to April), when ground temperature seldom falls below 0 ºC, was still higher than that in NC (Fig. 6). High soil creep velocity just after freeze-thaw seasons was also observed in a natural deciduous forest in the Ikawa University forest, where diurnal freeze-thaw is active because of leaf fall in the winter season (Imaizumi et al., 2017). High soil creep velocity in early spring is likely due to destruction of the soil structure caused by frequent freeze-thaw cycles in winter (Regüés and Gallart, 2004; Kværnø and Øygarden, 2006; McCool et al., 2013).

In seasons without freezing-thawing cycles, soil creep was identified only during heavy rainfall events, except early spring (Figs. 8, 9). Although rainfall (throughfall) intensity in site CC was higher than that in site NC (Fig. 7), the difference in rainfall thresholds for large soil creep velocity (> 2 mm day$^{-1}$) was not clear between the two sites (Fig. 9). The ratio of rainfall intercepted by forest canopies is low when total rainfall depth and rainfall intensity is high (Xiao et al., 2000; Fan et al., 2014). Such low interception rates during heavy rainfall events, when soil creep is active in the study site, obscured the difference in the rainfall threshold between the two sites.

### 5.4 Forest harvesting impact on the sediment flux

Winter sediment flux in site CC clearly decreased after forest harvesting in spite of the high soil creep velocity (Fig. 6 and Table 4). One reason for this trend is the decrease in freeze and creep thickness after forest harvesting (Fig. 5). In addition, previous studies emphasized that sediment transport activity and erosion rate in small plots are different from that on hillslope and catchment scales, because of the discontinuity of the sediment transport on hillslopes (Moreno-de las Heras et al., 2010; Sidle et al., 2017). By the field survey in CC, we identified sediment captured by the branches of harvested trees, especially where branches were piled up by forestry operations. Previous studies also reported that litters and woody debris on the ground prevented sediment transport (Hartanto et al., 2003; Liu et al., 2017). Thereby, interruption of sediment transport by branches resulted in differences in sediment transport activity between the micro and the hillslope scales, as monitored by TLCs and sediment traps, respectively.




Following forest harvesting, sediment flux decreased during rainfall periods in site CC, while differences in the sediment flux in NC before and after harvesting were not significant (Table 4). Decreases in the sediment flux of fine sediment, which is selectively transported by overland flow (Heng et al., 2011; Zhao et al., 2014), was clear during the rainfall seasons (Fig. 11). In site CC, the percentage of ground covered by understories, which reduces kinetic energy of raindrops splashing soil particles (Fukuyama et al., 2010; Nanko et al., 2015), changed from < 20% to > 90% within one year of forest harvesting (Fig. 1). In addition, the forest canopy, which increases the size of raindrop thus increasing their kinetic energy (Nanko et al., 2015), were removed by forest harvesting. Therefore, decreases in the sediment flux during the rainfall seasons was due to decreases in surface erosion rates, together with the capture of sediment by branches and litters of harvested trees (Fig. 13).

(a) Before forest harvesting

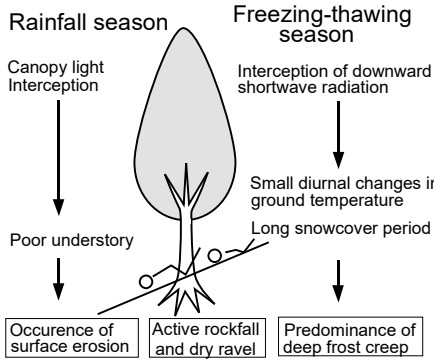

(b) After forest harvesting

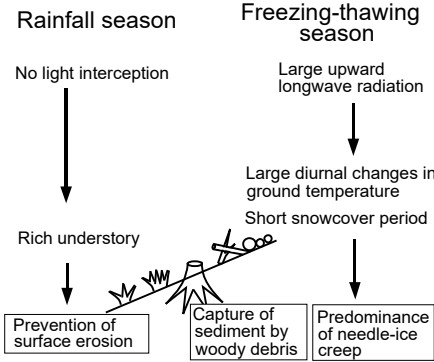

Figure 13: Difference in the hydrogeomorphic processes before and after forest harvesting on steep slopes in humid periglacial areas.





### 6 Summary and conclusion

The impact of forest harvesting on the hydrogeomorphic processes in the humid periglacial environment was investigated by intensive and comprehensive field observation of micrometeorological conditions and sediment transport. Harvesting of artificial conifer forests clearly changed micrometeorological conditions and sediment transport activities on a small scale (< 1 m2), especially during the winter season. Canopy removal increased the diurnal amplitude of net radiation and ground temperature, and also decreased the duration of snow cover. Such changes shortened freeze-thaw cycles from seasonal (e.g., for one week to half a month) to daily and made the freezing depth shallower. As a result, types of soil creep changed from long-lasting frost creep to daily and frequent needle-ice creep. The velocity of ground surface sediment during winter and spring in harvested areas was higher than non-harvested areas due to high needle-ice creep activity. Frequency of soil creep in the rainfall seasons (late spring to autumn) was not significantly different in harvested and non-harvested sites. Sediment flux clearly decreased both in freeze-thaw and rainfall seasons. Branches of harvested trees interrupted continuous sediment transport on hillslopes. In addition, growth of understories after forest harvesting reduced surface erosion by rain splash.

Our study clarified that forest harvesting both promotes and restrains sediment transport activity. Promoting factors are changes in the micrometeorological conditions by removal of the forest canopy, such as increases in the diurnal range of ground temperature, shortening of snow cover period, and increases in the throughfall. Restraining factors are the trap of sediment by branches of harvested trees and the growth of understories. The effect of forest harvesting on sediment flux would be variable among regions, depending on the climate and forest harvesting methods used, which control the impacts of each factor.

### Acknowledgement

The monitoring site and some of the rainfall data used in this study was provided by the Ikawa University Forest, University of Tsukuba. We thank the technical staff of the Ikawa University Forest, Toru Endo, Akira Takinami, Yoshikazu Endo, and Yusuke Ueji who supported our fieldwork. We are also grateful to Daiki Nosaka and other students in Shizuoka University, who conducted fieldwork with us.

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
