# Peer review of "Forest harvesting impacts on microclimate conditions and sediment transport activities in a humid periglacial environment"

_Hydrology and Earth System Sciences, 2018_

## Referee Comment (RC1) · Anonymous Referee #1 · 26 Mar 2018

In general, this study is a very interesting. The article has a clear idea. The research method is reasonable, the content is detailed and data is reliable. However, there are still some places that need to be revised, and I will mention it and suggest that the author supplement it. So I suggested that this manuscript should be published after minor modifications. 1.I suggest that the author add "Sediment transport activities in the periglacial environment are controlled by hillslopes micrometeorological conditions (i.e., air and ground temperatures, ground water content), which are highly affected by vegetation cover. Thus, there is a possibility that forest harvesting, which is the most dramatic change to vegetation cover in mountain areas, may severely impact sediment transport activities in periglacial areas (i.e., soil creep, dry ravel). Knowledge

of the effects of forest harvesting on sediment transport are needed to protect aquatic ecosystems as well as to develop better mitigation measures for preventing sediment disasters." in abstract part into introduction. 2.I suggest that the author add the main conclusions (including specific change indicators) into the abstract part. 3.Although various changes have been put forward before and after forest harvesting in this paper, there is no specific quantitative index and data explanation. 4.There are a lot of pictures in this article, but some of the graphs are a little messy. I suggest the author revise the picture. When the reader sees the picture, they will understand the scientific meaning of this picture. 5.The conclusion part is only of a list of the results. I recommend its refining. 6.There are many problems in the language of this article. Please modify carefully.

---

## Author Comment (AC1) · 12 Apr 2018

We sincerely thank you for the efforts you have made to improve our paper submitted to Hydrology and Earth System Sciences. We have responded to all review comments in the following paragraphs.

[Comment] In general, this study is a very interesting. The article has a clear idea. The research method is reasonable, the content is detailed and data is reliable. However, there are still some places that need to be revised, and I will mention it and suggest that the author supplement it. So I suggested that this manuscript should be published after minor modifications.

[Figure]

[Reply] It is our great pleasure that the reviewer is interested in our study. We think comments from the reviewer are very helpful for us to improve our manuscript. Please see replies listed below.

[Comment] 1. I suggest that the author add "Sediment transport activities in the periglacial environment are controlled by hillslopes micrometeorological conditions (i.e., air and ground temperatures, ground water content), which are highly affected by vegetation cover. Thus, there is a possibility that forest harvesting, which is the most dramatic change to vegetation cover in mountain areas, may severely impact sediment transport activities in periglacial areas (i.e., soil creep, dry ravel). Knowledge of the effects of forest harvesting on sediment transport are needed to protect aquatic ecosystems as well as to develop better mitigation measures for preventing sediment disasters." in abstract part into introduction.

[Reply] Thank you for your suggestion. We think the section is the key point of our paper. We will insert the section into introduction to emphasize the key point of our paper.

[Comment] 2. I suggest that the author add the main conclusions (including specific change indicators) into the abstract part.

[Reply] Based on the comments on by the reviewer, we will add the main conclusions of this paper in the abstract.

[Comment] 3. Although various changes have been put forward before and after forest harvesting in this paper, there is no specific quantitative index and data explanation.

[Reply] As the reviewer points out, many changes in micrometeorological conditions and sediment transport processes are qualitatively explained in the text. We will add quantitative explanations in the text. For example, we will improve statements on net radiation (pg. 10, lines 2-4), frequency of freeze-thaw cycle (pg. 10, lines 11-12), and snow depth (pg. 10, lines 13-16) by showing specific values.

[Comment]4. There are a lot of pictures in this article, but some of the graphs are a little messy. I suggest the author revise the picture. When the reader sees the picture, they will understand the scientific meaning of this picture.

[Reply] We think the reviewer concerns about Figs. 5 and 11, which include many graphs inside. We will separate graphs in Figs. 5 into two different figures. We will divide Fig. 11 into two figures, too.

[Comment] 5. The conclusion part is only of a list of the results. I recommend its refining.

[Reply] We will simplify the first paragraph in the "Summary and Conclusion", in which results and discussion in this paper are summarized. In addition, we will add conclusive sentences in the conclusion.

[Comment] 6. There are many problems in the language of this article. Please modify carefully.

[Reply] We will ask a native English speaker to check English throughout the manuscript.

Thank you again for your helpful comments.

---

## Referee Comment (RC2) · Anonymous Referee #2 · 19 May 2018

General comments: This study carried a careful field experiment for studying forest harvesting impacts on micrometeorological conditions and sediment transport activities in a humid periglacial environment. It is important for management of the periglacial catchment, especially the vegetation-erosion processes. The observation methods were generally reliable, the datasets showed good quality, and the presentation of results were also clear. However, the discussion section requires improvement as the present version is more or less repeating of the results rather than a discussion. The discussion should focus on showing a more general cognition that helps people understand micrometeorological conditions and sediment transport activities in a humid periglacial, and the influence of forest harvesting on such processes. In addition, the

abstract should be revised with less common sense but more scientific findings from this study.

Specific comments: There are several points require correction or clarification: Page 2 line 5: as this study is not relevant to aquatic ecosystems, I would suggest delete the sentence. Page 3 line 27: why not arrange the CC and NC at the same contours with similar slope gradient? As the steep slope is apt to failure, it is inappropriate to just neglect the influence of the different slopes. Please clarify! Page 4 Table 1: the difference of the contributing area would also affect the calculation of sediment yield, e.g. a smaller area would give a larger sediment yield rate. So the difference in the cross-sectional topography could not be distinguished from the comparison of the ridge, straight, and valley. Please clarify! Page 5 lines 10-14: it should be explained how to deal with the non-measured periods/ or why it is acceptable with such discontinuous measurement. Page 5 lines 15-20: as you have both temperature logger data and some short period radiometer data, why not try to correlate the two datasets and extension of the radiometer data? Figure 2: the high boulders at CCV acted as flow resistance structure and could reduce erosion ability of flow and may not be ignored, therefore the influence of vegetation clearance may not be distinguished by the comparison of CC and NC. Page 10 lines 18-19, the 0-3 mm hr-1 difference in rainfall intensity between the CC and NC is not clearly seen from Fig. 7. Figure 4: typing error of "(b) after harvesting" Figure 8: why CC not measured for the sampling period as NC? for the different peaking rainfall intensities, how the velocities of CC and NC along slope were comparable? Figure 9: the uncertainty should be indicated as there is one dot of NC having no clear displacement of ground surface sediment at the maximum hourly rainfall as high as 11 mm hr-1 Figure 11: typing errors in the caption, see (g), (i), (j) Page 23 lines 13-16: I would suggest write the sentences as "Our study clarified that forest harvesting promoted changes in the micrometeorological conditions by removal of the forest canopy, such as increases in the diurnal range of ground temperature, shortening of snow cover period, and increases in the throughfall. However, sediment transport activity has been restrained due to the trap of sediment by branches

of harvested trees and the growth of understories."

---

## Author Comment (AC2) · 7 Jun 2018

We sincerely thank you for the efforts you have made to improve our paper submitted to Hydrology and Earth System Sciences. We have responded to all review comments in the following paragraphs.

<General Comments> This study carried a careful field experiment for studying forest harvesting impacts on micrometeorological conditions and sediment transport activities in a humid periglacial environment. It is important for management of the periglacial catchment, especially the vegetation-erosion processes. The observation methods were generally reliable, the datasets showed good quality, and the presentation of re-

sults were also clear. However, the discussion section requires improvement as the present version is more or less repeating of the results rather than a discussion. The discussion should focus on showing a more general cognition that helps people understand micrometeorological conditions and sediment transport activities in a humid periglacial, and the influence of forest harvesting on such processes. In addition, the abstract should be revised with less common sense but more scientific findings from this study.

[Reply] It is our pleasure that the reviewer understand importance of our study. Based on comments from the reviewer, we will remove sentences repeating results in discussion section. In addition, we will add general findings about forest harvesting impacts on micrometeorological conditions and sediment transport activities. In the abstract, we will replace ambiguous expressions with the scientific explanations.

<Specific comments> [Comment] 2 line 5: as this study is not relevant to aquatic ecosystems, I would suggest delete the sentence.

[Reply] We will remove the sentence as suggested by the reviewer.

[Comment] Page 3 line 27: why not arrange the CC and NC at the same contours with similar slope gradient? As the steep slope is apt to failure, it is inappropriate to just neglect the influence of the different slopes. Please clarify!

[Reply] The harvesting area was decided by conditions of trees, access to the area, and ease of logging. Our study was not most important criterial for the decision of harvesting area. Although we tried to select control sites (NC) with similar tree conditions and topography as possible, topography in NC was slightly different from that in CC. We will add potential effects of the topography on differences in sediment transport activity between CC and NC.

[Comment] Page 4 Table 1: the difference of the contributing area would also affect the calculation of sediment yield, e.g. a smaller area would give a larger sediment yield

rate. So the difference in the cross-sectional topography could not be distinguished from the comparison of the ridge, straight, and valley. Please clarify!

[Reply] We did not divide the sediment transport rate by the contributing area to avoid scaling effect pointed out by the reviewer. The sediment transport rate was divided by width of sediment traps to obtain sediment flux. We will add an explanation on this point.

[Comment] Page 5 lines 10-14: it should be explained how to deal with the non-measured periods/ or why it is acceptable with such discontinuous measurement.

[Reply] Throughfall was just analyzed in the Fig. 7. Periods without data are not shown in the figure. Because of the intermission of throughfall monitoring, throughfall was not used in the analysis of sediment transport rate (e.g., Fig. 9). We will explain how throughfall data was used in this study.

[Comment] Page 5 lines 15-20: as you have both temperature logger data and some short period radiometer data, why not try to correlate the two datasets and extension of the radiometer data?

[Reply] Thank you for your helpful comment. We will try to find relationship between temperature logger data and radiometer data, and extend the radiometer data using the relationship if possible.

[Comment] Figure 2: the high boulders at CCV acted as flow resistance structure and could reduce erosion ability of flow and may not be ignored, therefore the influence of vegetation clearance may not be distinguished by the comparison of CC and NC.

[Reply] We agree that the flow resistance in CCV is likely higher than the other plots because of the large boulder size. Therefore, impact of forest harvesting cannot be simply discussed by comparison of the data in CC and NC. Difference in the topography between CC and NC, which are pointed out by the reviewer, also affects sediment flux. In this study, the forest harvesting impact was discussed based on the comparison of

sediment flux before and after the harvesting in each site (Figs. 10, 11, 12, table 2). We will explain this point in the first part of the paper.

[Comment] Page 10 lines 18-19, the 0-3 mm hr-1 difference in rainfall intensity between the CC and NC is not clearly seen from Fig. 7.

[Reply] In Fig. 7, we will shade the range (0-3 mm hr-1 higher than x=y) in order to clarify the trend. In addition, we calculated total duration that rainfall intensity in CC exceeded that in NC. Ratio of the duration (CC > NC) in the total rainfall period was 0.50 and 0.62 before and after the harvesting, respectively. We will note that in the text.

[Comment] Figure 4: typing error of "(b) after harvesting"

[Reply] Thank you. We will revise the spelling.

[Comment] Figure 8: why CC not measured for the sampling period as NC? for the different peaking rainfall intensities, how the velocities of CC and NC along slope were comparable?

[Reply] The periods when TLCs worked both in CC and NC are limited because of the mechanical troubles. Additionally, heavy rainfall events were not observed in such periods. Therefore, we could not show the figure with same sampling period. We will note that in the manuscript. As the reviewer points out, the velocity in NC and CC cannot be simply compared because of different rainfall intensities. Therefore, we did not compare soil creep velocity in Fig. 8. Alternatively, we compared the velocity in Fig. 9.

[Comment] Figure 9: the uncertainty should be indicated as there is one dot of NC having no clear displacement of ground surface sediment at the maximum hourly rainfall as high as 11 mm hr-1

[Reply] As the reviewer points out, the relationship includes some uncertainty. We will note that in the text.

[Comment] Figure 11: typing errors in the caption, see (g), (i), (j)

[Reply] We will replace COR and COS with NCR and NCS.

[Comment] Page 23 lines 13-16: I would suggest write the sentences as "Our study clarified that forest harvesting promoted changes in the micrometeorological conditions by removal of the forest canopy, such as increases in the diurnal range of ground temperature, shortening of snow cover period, and increases in the throughfall. However, sediment transport activity has been restrained due to the trap of sediment by branches of harvested trees and the growth of understories."

[Reply] Thank you for your suggestion. We think the suggestion is better expression as conclusion of the paper. We will revise the sentence as suggested by the reviewer.

Thank you again for your helpful comments.

---

## Author Response (AR1)

**Reply for review comments**

We sincerely thank you for the efforts you have made to improve our paper submitted to *Hydrology and Earth System Sciences*. We have responded to all review comments in the following paragraphs. The blue-highlighted sentences are the review comments; sentences in black represent our responses to these review comments.

**Reviewer 1**

[Comment] In general, this study is a very interesting. The article has a clear idea. The research method is reasonable, the content is detailed and data is reliable. However, there are still some places that need to be revised, and I will mention it and suggest that the author supplement it. So I suggested that this manuscript should be published after minor modifications.

[Reply] It is our great pleasure that the reviewer is interested in our study. We have revised our manuscript based on helpful comments from the reviewer. Please see replies listed below.

[Comment] 1. I suggest that the author add "Sediment transport activities in the periglacial environment are controlled by hillslopes micrometeorological conditions (i.e., air and ground temperatures, ground water content), which are highly affected by vegetation cover. Thus, there is a possibility that forest harvesting, which is the most dramatic change to vegetation cover in mountain areas, may severely impact sediment transport activities in periglacial areas (i.e., soil creep, dry ravel). Knowledge of the effects of forest harvesting on sediment transport are needed to protect aquatic ecosystems as well as to develop better mitigation measures for preventing sediment disasters." in abstract part into introduction.

[Reply] Thank you for your suggestion. We think the section is the key point of our paper. We have inserted the section at the top of introduction to emphasize the key point of our paper (pg. 2, lines 2-7).

[Comment] 2. I suggest that the author add the main conclusions (including specific change indicators) into the abstract part.

[Reply] We think the conclusions in the previous version of the abstract was not clear. Based on the comment by reviewer, we have added name of specific indicators, which are impacted by the forest harvesting, in the conclusion (pg. 1, lines, 27-29). Additionally, we have added quantitative explanations on the changes in the micrometeorological conditions and sediment transport activity in the abstract (pg. 1, lines 21-24).

[Comment] 3. Although various changes have been put forward before and after forest harvesting in this paper, there is no specific quantitative index and data explanation.

[Reply] As the reviewer points out, many changes in micrometeorological conditions and sediment transport processes were qualitatively explained in the previous version of the manuscript. Based on this comment, we have added quantitative explanations in the text. For example, we have improved statements

on ground surface temperature (pg. 10, 2-6), net radiation (pg. 10, lines 6-9), snow depth (pg. 10, lines 10-12), frequency of freeze-thaw cycle (pg. 10, lines 14-21), soil creep velocity (pg. 14, line 16 - pg. 15, line 2) by showing specific values.

[Comment 4] There are a lot of pictures in this article, but some of the graphs are a little messy. I suggest the author revise the picture. When the reader sees the picture, they will understand the scientific meaning of this picture.

[Reply] We think the reviewer concerns about Figs. 5 and 11 (in previous version of the manuscript), which included many graphs inside. We have separated previous Figs. 5 into two different figures (new Figs. 5 and 6). New Fig. 5 shows effect of radiation and snow cover on the ground surface temperature. New Fig. 6 compares temporal changes in the ground temperature and frost heave activities. We also divided previous Fig. 11 into two figures (Figs 13, 14), too.

[Comment 5] The conclusion part is only of a list of the results. I recommend its refining.

[Reply] We will simplify the first paragraph in the "Summary and Conclusion", in which results and discussion in this paper are summarized. In addition, we have added conclusive sentences (pg. 24, lines 7-16)

[Comment 6] There are many problems in the language of this article. Please modify carefully.

[Reply] A native English speaker have edited English in this manuscript again. We think the language has been improved by the editing.

Thank you again for your helpful comments.

We sincerely thank you for the efforts you have made to improve our paper submitted to *Hydrology and Earth System Sciences*. We have responded to all review comments in the following paragraphs.

<General Comments> This study carried a careful field experiment for studying forest harvesting impacts on micrometeorological conditions and sediment transport activities in a humid periglacial environment. It is important for management of the periglacial catchment, especially the vegetation-erosion processes. The observation methods were generally reliable, the datasets showed good quality, and the presentation of results were also clear. However, the discussion section requires improvement as the present version is more or less repeating of the results rather than a discussion. The discussion should focus on showing a more general cognition that helps people understand micrometeorological conditions and sediment transport activities in a humid periglacial, and the influence of forest harvesting on such processes. In addition, the abstract should be revised with less common sense but more scientific findings from this study.

[Reply] It is our pleasure that the reviewer understand importance of our study. We think the last part of this comment is very important to improve our manuscript. Based on this comment, we have revised throughout the discussion. We have reduced sentences that explain specific results in the Ikawa University forest (e.g., pg. 20, lines 18-21; pg. 21, line 27-28; pg. 21 line 32; pg. 22, line 4; pg. 22, line 5-7). Instead, we have added sentences on the general findings (pg. 20, lines 16-17; pg. 21, line 27-28; pg. 22, line 14-15).

In the abstract, we have replace ambiguous expressions with the scientific explanations which includes name of specific indicators and quantitative values.

<Specific comments>

[Comment] 2 line 5: as this study is not relevant to aquatic ecosystems, I would suggest delete the sentence.

[Reply] We have remove the sentence as suggested by the reviewer (pg. 2 line 2-8).

[Comment] Page 3 line 27: why not arrange the CC and NC at the same contours with similar slope gradient? As the steep slope is apt to failure, it is inappropriate to just neglect the influence of the different slopes. Please clarify!

[Reply] The harvesting area was decided by forest manager (Tsukuba University) based on conditions of trees, access to the area, and ease of logging. Our study was not most important criterial for the decision of harvesting area. Although we tried to select control sites (NC) with similar tree conditions and topography as possible (pg. 4, lines 3-5), slope gradient in NC was slightly lower than that in CC. We have add potential effects of the different topography on monitoring results (pg. 9, line 8-pg. 10 line 2; pg.16, lines 2-7).

[Comment] Page 4 Table 1: the difference of the contributing area would also affect the calculation of

sediment yield, e.g. a smaller area would give a larger sediment yield rate. So the difference in the cross-sectional topography could not be distinguished from the comparison of the ridge, straight, and valley. Please clarify!

[Reply] We did not divide the sediment transport rate by the contributing area to avoid scaling effect as pointed out by the reviewer (Table 4). The sediment transport rate was divided by width of sediment traps to obtain sediment flux. The sediment transport rate is generally higher at the plots with larger contributing area (Table 4). We have added explanations on this point (pg. 8, lines 25-26; pg. 16, lines 1-7).

[Comment] Page 5 lines 10-14: it should be explained how to deal with the non-measured periods/ or why it is acceptable with such discontinuous measurement.

[Reply] Throughfall was just analyzed in the Fig. 7. Periods without data are not shown in the figure. Throughfall was not used in the analysis of sediment flux (e.g., Fig. 10,) because of the intermission of throughfall monitoring. We will explain how throughfall data was used in this study (pg. 5, lines 14-15).

[Comment] Page 5 lines 15-20: as you have both temperature logger data and some short period radiometer data, why not try to correlate the two datasets and extension of the radiometer data?

[Reply] Thank you for your helpful comment. As suggested by the reviewer, we found relationship between temperature logger data and radiometer data, and extend the radiometer data using the relationship (pg. 5, lines 19-24).

[Comment] Figure 2: the high boulders at CCV acted as flow resistance structure and could reduce erosion ability of flow and may not be ignored, therefore the influence of vegetation clearance may not be distinguished by the comparison of CC and NC.

[Reply] We agree that the flow resistance in CCV is likely higher than the other plots because of the large boulder size. Therefore, impact of forest harvesting cannot be simply discussed by comparison of the data between CC and NC. Difference in the topography between CC and NC, which are pointed out by the reviewer, also affects sediment flux. In this study, the forest harvesting impact was discussed based on the comparison of sediment flux before and after the harvesting in each site (Figs. 11, 12, 13, 14, Table 4). Comparison of sediment flux between CCV and other site was not used to evaluate the impact. We have explained this point in the first part of the paper (pg. 8, lines 26-32).

[Comment] Page 10 lines 18-19, the 0-3 mm $hr^{-1}$ difference in rainfall intensity between the CC and NC is not clearly seen from Fig. 7.

[Reply] In Fig. 8, we shaded the range (0-3 mm $hr^{-1}$ higher than x=y) in order to clarify the trend. In addition, we calculated total duration that rainfall intensity in CC exceeded that in NC. Ratio of the duration (CC > NC) in the total rainfall period was 0.50 and 0.62 before and after the harvesting, respectively. We will note that in the text (pg. 10, line 27-28).

[Comment] Figure 4: typing error of "(b) after harvesting"

[Reply] Thank you. We have revised the spelling.

[Comment] Figure 8: why CC not measured for the sampling period as NC? for the different peaking rainfall intensities, how the velocities of CC and NC along slope were comparable?

[Reply] The periods when TLCs worked both in CC and NC are limited because of the mechanical troubles. Additionally, heavy rainfall events were not observed in such periods. Therefore, we could not show the figure with same sampling period. We have noted that in the Figure caption (Fig. 9). As the reviewer points out, the velocity in NC and CC cannot be simply compared because of the different rainfall intensity. Therefore, we did not compare soil creep velocity in Fig. 9. Alternatively, we compared the velocity shown in Fig. 10.

[Comment] Figure 9: the uncertainty should be indicated as there is one dot of NC having no clear displacement of ground surface sediment at the maximum hourly rainfall as high as 11 mm hr-1

[Reply] As the reviewer points out, the relationship includes some uncertainty. We have indicated it in the text (pg. 15, lines 6-7).

[Comment] Figure 11: typing errors in the caption, see (g), (i), (j)

[Reply] We have replaced COR and COS with NCR and NCS (Fig. 13).

[Comment] Page 23 lines 13-16: I would suggest write the sentences as "Our study clarified that forest harvesting promoted changes in the micrometeorological conditions by removal of the forest canopy, such as increases in the diurnal range of ground temperature, shortening of snow cover period, and increases in the throughfall. However, sediment transport activity has been restrained due to the trap of sediment by branches of harvested trees and the growth of understories."

[Reply] Thank you for your suggestion. We think the suggestion is better expression as conclusion of the paper. We will revise the sentence as suggested by the reviewer (pg. 24, lines 7-10).

Thank you again for your helpful comments.

[revised manuscript text omitted]